# IDARE2—Simultaneous Visualisation of Multiomics Data in Cytoscape

**DOI:** 10.3390/metabo11050300

**Published:** 2021-05-06

**Authors:** Thomas Pfau, Mafalda Galhardo, Jake Lin, Thomas Sauter

**Affiliations:** 1Department of Life Sciences and Medicine, University of Luxembourg, 4365 Esch-sur-Alzette, Luxembourg; ThomasPfau@gmx.de (T.P.); galhardo.mafalda@gmail.com (M.G.); 2VDR Lab, Institute for Research and Innovation in Health (i3S), University of Porto, 4200-135 Porto, Portugal; 3Statistical and Population Genetics, Institute for Molecular Medicine Finland, University of Helsinki, 00100 Helsinki, Finland; Jake.Lin@helsinki.fi

**Keywords:** data visualisation, metabolic networks, network structure

## Abstract

Visual integration of experimental data in metabolic networks is an important step to understanding their meaning. As genome-scale metabolic networks reach several thousand reactions, the task becomes more difficult and less revealing. While databases like KEGG and BioCyc provide curated pathways that allow a navigation of the metabolic landscape of an organism, it is rather laborious to map data directly onto those pathways. There are programs available using these kind of databases as a source for visualization; however, these programs are then restricted to the pathways available in the database. Here, we present IDARE2 a cytoscape plugin that allows the visualization of multiomics data in cytoscape in a user-friendly way. It further provides tools to disentangle highly connected network structures based on common properties of nodes and retains structural links between the generated subnetworks, offering a straightforward way to traverse the splitted network. The tool is extensible, allowing the implementation of specialised representations and data format parsers. We present the automated reproduction of the original IDARE nodes using our tool and show examples of other data being mapped on a network of *E. coli*. The extensibility is demonstrated with two plugins that are available on github. IDARE2 provides an intuitive way to visualise data from multiple sources and allows one to disentangle the often complex network structure in large networks using predefined properties of the network nodes.

## 1. Introduction

With the ever increasing number of “omics” data, it is becoming increasingly important to handle and combine multiple types of data and interpret the experimental findings. While there are plenty of statistical approaches to interpret omics data, those approaches tend to provide a set of isolated targets. Placing those targets into context and visualizing the measurements are keys to obtain ideas about the effects of a given treatment. There are many tools dedicated to visualization of omics data in a network context (see, e.g., the review by Gehlenborg et al. [1]), and tools like Paintomics [2,3], BiNA [4] or MONGKIE [5] have started to address the issue of simultaneous multi-omics visualisation with recent tools like MetaboMAPS [6] or SAMMI [7], giving increasingly better and more satisfying results. Escher [8] and Pathway Tools [9] are popular applications tailored for building and sharing biological pathway and metabolic visualizations across multiple model organisms. In addition, Fluxer [10] has been designed to impressively compute and visualize genome-scale metabolic flux networks. One feature often lacking in the available tools is the ability to generate custom processing code for repetitive analyses or more complex data handling, with exceptions such as Omix [11], which, however, necessitates the use of its OVL scripting language.

Among the general-purpose network visualization tools, Cytoscape [12] has become prominent as it provides a well-documented API (Application Programming Interface), is open-source and offers an ever increasing number of apps addressing different problems in its app store [13]. In Cytoscape, several applications that aim at handling data exist [14,15]. MetScape [14], which is able to visualize metabolomics and transcriptomics data and is tailored for metabolic networks, or the general enhanced Graphics app [15], which allows the generation of multiple different types of visualization with small scripts. While these tools do allow visualization in an integrated way, they are either restricted in the usable data (e.g., MetScape handles expression or metabolomics data but does not support proteomics or simulated data) or necessitate additional scripting by the user to adjust the visualization (like the enhancedGraphics app).

However, to our knowledge, no Cytoscape app exists that allows the visualization of data from multiple sources using a graphical user interface that is easily usable by all users and works with the current Cytoscape version. Another issue often encountered in network analysis is the problem of untraceability of edges, due to a large network size and high connectivities. This is particularly true in metabolic networks, where common metabolites (like water or protons) tend to induce a hairball structure during layout making visual inspection tiresome. A possible way to address this is the separation of the network into pathways or subnetworks. However, investigation of this reduced view is prone to missing important connections between the currently investigated pathway and the remaining network. In databases such as KEGG [16] or the BioCyc [17], which are based on Pathway Tools [9], this is commonly addressed by introducing links between the pathways at important interaction points. For Cytoscape, Cerebral [18] tried to offer similar functionality but was restricted to a single network view and thus generates many overlapping edges between the different subnetworks.

With the Integrated DAtanodes of REgulation (IDARE) [19], we introduced an approach to combine numerous sources of information and visualize them in the context of metabolic networks. The concept allows the simultaneous interpretation of multiple experiments and simulations in a network context and thus provides an integrative method of visual inspection.

Here we present IDARE2, a Cytoscape app that addresses both the general visualization and the network layout challenge. The IDARE2 app provides the concept of multiomics visualization introduced in IDARE in an automated fashion, not restricted by the type of underlying network. Instead of manual image generation outside of cytoscape, which was necessary in IDARE, the app allows the automated generation of multiomics data visualizations within the Cytoscape framework. The app allows the incorporation of multiple datasets originating from diverse experimental setups or simulations and select those that shall be mapped to the nodes in a network. It allows the addition of these images to any Cytoscape network visualization and provides an explanatory legend for each image based on the used data. The app further addresses the issue of an emergent hairball structure by generating subnetworks based on properties of the nodes and links them, generating linker nodes similar to those mentioned above. These links allow an intuitive navigation through the different pathways of a complex network, while the separation removes overlapping edges, which otherwise make inspection tedious. Finally, the app provides a Systems Biology Markup Language (SBML) annotation reader that, among other things, can extract available pathway information from the SBML file. This information is often available in SBML representations of metabolic networks and makes subnetwork generation easier. Our app therefore presents a versatile approach that facilitates the investigation of complex networks and data in a visual way while keeping the network context.

## 2. Results

We created a multiomics visualization platform integrated into the Cytoscape framework. The IDARE2 app provides multiple visualization and dataset types and offers support for individual, time series, line graph, scatter plot and heat map visualization styles. (Details and a user guide can be found on https://github.com/sysbiolux/IDARE accessed on 24 April 2021). While it is straightforward to generate nodes representing the data used in the original IDARE paper [19] (e.g., the activation of the Triacyl glycerol synthesis pathway with automatically generated nodes seen in Figure 1), the new tool can also be applied to other data without effort. Loading the data can easily be achieved from within Cytoscape, and different types of data can be provided in excel or comma separated value formats. The layout is done automatically and generates both nodes and legends for the generated nodes. The app further allows modification of the automated layout if the automatic alignment does not meet the requirements or if more complex alignments are needed.

The tool allows the simultaneous generation of images with data specific to different nodes, leaving the unknown information blank for those nodes where the data are not available. This allows the tool to achieve a consistent data representation for all nodes for which any data are available. Alternatively, nodes can be created for multiple subsets of the available data, thus generating different images for different types of nodes. To distinguish between different node types, IDARE2 generates legends for each node type and generates a node-legend mapping for each node. The legends also provide details about each supplied dataset to allow a clear distinction between similar datasets. The interface provided by the application is displayed in Figure 2, which shows visualized data from the example provided on github on a network of *Escherichia coli*. The generated nodes can be mapped to any Cytoscape network that contains the identifiers provided in the data. By providing node label information, alternate IDs can be displayed on the image nodes. The visualisation is independent of the model used, as long as the IDs in the data match the IDs in the model. IDARE2 provides interfaces for users to generate their own visualization styles to display on nodes, and to create their own parsers and dataset types. This makes the app flexible and allows to implement specific requests without changing the underlying app. All generated nodes and legends can be exported to SVG or PNG graphics, making them easy to use in publications or network independent interpretation.

### 2.1. Subnetwork Generation

Complex network structures, especially in highly connected networks, often result in a hairball structure of the laid out network. While tools like Cerebral [18] allow the distribution of a network into compartment or pathway layers, those layers might not be hierarchic, which can lead to numerous trans-compartment edges, leaving problems with visual inspection. In addition, there is currently no application that provides similar services for Cytoscape 3.x, as Cerebral is incompatible with the current Cytoscape version. A simple splitting of networks based on node properties can be easily achieved with the default Cytoscape tools. However, this type of split needs the properties to be present on all nodes that belong to a certain subsystem. For metabolic networks, or other bipartite networks, this is commonly not true, since only objects of one class have a specific property, which indicates their association with a subnetwork. In metabolic networks, for example, one would not only like to extract the reactions, but also the connecting metabolites and potentially any linked genes or protein nodes. Thus, the generation of subnetworks that contain all nodes associated with a given property is desirable. However, if such a subnetwork is disconnected from the remaining network, trying to determine distributions of flux over the whole network can become inconvenient. This is a problem addressed in KEGG [16] or MetaCyc [17] by introducing links to other subnetworks (i.e., pathways). The new IDARE2 app allows the user to generate extended subnetworks for bipartite networks, adding links between networks that are easy to follow. These links allow an intuitive navigation between the subnetworks, thus retaining the connectivity of the network while simultaneously making the network more tractable. The user can select the node types that form the bi-partition (e.g., species and reactions) and can decide which type belongs to the subnetwork, and which type bridges between subnetworks. Finally, the user can select which nodes to exclude from the subnetworks, and whether a set of nodes should not be used to link between subsystems. In metabolic networks removed nodes commonly include abundant metabolites like water or protons, while non-branching nodes contain less abundant co-factors like Coenzyme A. IDARE2 suggests sets for both removed and non linking nodes based on their connectivity in the network. An example of generated subnetworks would be the compartments of the metabolic network, where species belong to the compartment and reactions can link between them. The generated subnetworks could then be further split into pathways, where reactions are parts of the pathways and metabolites can be used as links. An example of glycolysis/gluconeogenesis, the citric acid cycle and the pentose phosphate pathway in the *E. coli* core network [20] is provided in Figure 3. Depending on the user’s choice, a pathway can be split into different compartments or will have breaks if transport reactions are not part of the pathway. However, links to the transport subsystem would be created, indicating these connections.

### 2.2. Duplication of Highly Connected Nodes

As mentioned above, highly connected co-factors make automatic layout extremely challenging in larger metabolic networks. Unfortunately, Cytoscape does not yet offer an option to easily create special nodes representing replicates of a co-factor node but being connected, such that updating properties of one node automatically updates the other nodes. IDARE2 offers this type of linked node replication. The nodes can also be remerged if required and are not considered as branching for subnetwork generation purposes.

### 2.3. SBML Annotation for Metabolic Networks

SBML files commonly contain substantial annotations for a given network. While tools like CySBML [21] provide access to these annotations, there are some annotations, particularly in metabolic networks, that are often stored in the notes section (see the COnstraint Based Reconstruction and Analysis (COBRA) definitions in [22]). These COBRA annotations contain information about gene associations or pathways associated with specific reactions. CySBML displays parts of this information when encountering it but does not make it directly accessible to the user via table columns. Another way to encode gene protein reaction associations is to use species as enzymes and modifier species to assign these enzymes to reactions (see [23]). In both instances, the genes are commonly not visible as nodes in the network using standard Cytoscape features, which would be preferable for visualization of gene-specific data. IDARE2’s SBML annotator can extract this information from the SBML file (either using the JSBML [24] library or the SBMLManager provided by Cy3SBML), to instantiate gene and protein nodes. The latter are already present in a network loaded with Cy3SBML. The annotator changes the SBML type of species that either exhibit a matching SBOTerm (0000014 for enzyme or 0000252 for polypeptide chain, see http://www.ebi.ac.uk/sbo/ accessed on 26 April 2021) or have a Bio-qualifier [25] is *EncodedBy* to protein and creates nodes for any annotated genes, linking them to the respective proteins. If genes are directly associated with reactions (as is done in COBRA annotations), the gene nodes are linked directly to the associated reactions (the latter can be seen in Figure 3).

### 2.4. Examples

We used the IDARE app to visualize data on a small sample study into the ethanol stress tolerance of *E. coli* to indicate potential uses of the platform. The network used was the *E. coli* core model as described in [26]. This example demonstrates the integration of multiple sources of omics data and visualization of time course data using the automated node layout and generation features of IDARE2. We collected data from four different studies, including a proteomics time course of ethanol treated cells [27], gene expression of control vs. treated cells [28], metabolite timecourses upon treatment with ethanol [29] and a timecourse of C13 labeling of several central carbon metabolites [30].

From the visualization of the TCA (tricarboxylic acid cycle; see Figure 4), it is easily visible that the expression and protein amounts are reduced when the cells are exposed to ethanol stress. At the same time, metabolite levels are slightly increasing, which partly compensates for this effect. The labeling data nicely illustrate that the tolerant strain is quicker in incorporating the labeled ethanol, and, in the network view, it suggests that the glyoxylate shunt is potentially used due to the excess acetyl-CoA produced during ethanol absorbtion. This is supported by increased protein levels in both isocitrate lyase and malate synthase when comparing between a tolerant and wild-type strain and could explain the low incorporation of labels in α-ketoglutarate. The latter could actually be derived from amino acids (e.g., glutamate) available in the Lysogeny broth minimal medium used by Goodarzi et al. [30]. This example shows the multitude of data that can be combined using IDARE2 and already yields interesting hypotheses that could be addressed in future studies. A guide to recreate these figures and another example using a genome scale model of zebrafish [31] are available in the Quickstart-Guide (https://github.com/sysbiolux/IDARE-Quickstart accessed on 26 April 2021).

## 3. Discussion

We present a novel tool for visualization of any type of omics data on Cytoscape networks in a user friendly fashion. While earlier options included the necessity of scripting (e.g., the enhancedGraphics app), our app allows an intuitive management of datasets to quickly create visual representations of diverse sets of information on a given network. Furthermore, the legend provided by our app makes it easy to identify which plot represents which data and allows a quick assessment of the visualised information. The subsystem generation methodology introduces a straightforward way to navigate complex networks and allows the disentanglement of the common hairball structure. We tested the subnetwork generation and image processing on metabolic networks of up to 10,000 nodes and 25,000 edges, with data available for 3400 nodes, and demonstrated its versatility and usability on a small example of *E. coli* under ethanol stress. Generating the images takes a few seconds for all images to be available, indicating a good scalability. Subnetworks for this network scale were generated within seconds if no additional layout generation was requested. The layout of large subnetworks (e.g., the cytosol compartment of a metabolic network) can take a significant amount of time, depending on the layout algorithm chosen. The extensibility of the app allows it to easily add additional visualization types or to include computation pipelines for datasets by a specific reader (as demonstrated with the different options of the GEOSoft reader plugin).

## 4. Materials and Methods

While the concept we introduced in Galhardo et al. [19] is in theory applicable to any type of data and any type of network, the initial method relied on manual layout and design. Images had to be created by external tools and had to be manually mapped onto nodes in a Cytoscape network. With IDARE2, we provide a Cytoscape app that makes this concept more accessible and allows the automatic generation of image nodes and their mapping to networks from within Cytoscape. All functionalities of the created app are described in detail in the results section.

### 4.1. Extensibility

IDARE2 provides its own plugin system, which can be used to add the ability to read and represent additional types of data or integrate additional modes of visualisation, thus making the app flexible for future development. We provide an API for development of these plugins, which contains a few convenience functions and classes and is available at https://sysbiolux.github.io/IDARE/API/ (accessed on 26 April 2021). Further, we provide two sample plugins for these functionalities. One plugin adds bar charts as a visualization type (BarChartPlugin, accessed on 26 April 2021), and the other plugin allows the parsing of simple omnibus format in text (SOFT) files from the gene expression omnibus (GEOSoft plugin, accessed 26.4.2021). Both plugins, along with the IDARE2 source code, can be found on github (https://github.com/SysBioLux/ accessed on 26 April 2021).

### 4.2. Availability and Requirements

The Cytoscape application can be downloaded from the Cytoscape app store or directly from the project page at: https://github.com/sysbiolux/IDARE/releases (accessed on 26 April 2021).The source code can be found at https://sysbiolux.github.io/IDARE (accessed on 26 April 2021) while a quickstart guide is available at https://github.com/sysbiolux/IDARE-Quickstart (accessed on 26 April 2021 at revision 6166f5a). The plugin API can be found at https://sysbiolux.github.io/IDARE/API/ (accessed on 26 April 2021) and on the Cytoscape App store and a user guide is available on the github repository. All code is implemented in java and available under a LGPL v3.0 license.

## 5. Conclusions

With IDARE2, we introduce an intuitive way to visualize multi-omics data from various sources on cytoscape networks. IDARE2 makes metabolic network analysis in Cytoscape easier, providing means to quickly add attractive metanode images reflecting complex data without breaking the network connectivity, generating connected subsystems for pathways of interest. Finally, by allowing individual data imports and visualizations to register with the app, experienced users can easily adapt the app to their needs. 

## Figures and Tables

**Figure 1 metabolites-11-00300-f001:**
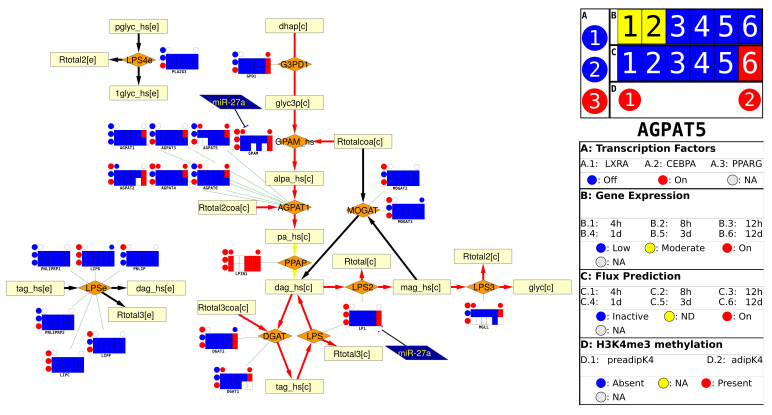
Data visualization of the tri-acyl glycerol synthesis pathway during differentiation of adipocytes. Activation of parts of the synthesis pathway at the end of the differentiation is easily visible. Reactions are represented as orange rhombus, metabolites by light yellow boxes and miRNAs as dark blue trapezoids. The metanode legend can be found on the right side.

**Figure 2 metabolites-11-00300-f002:**
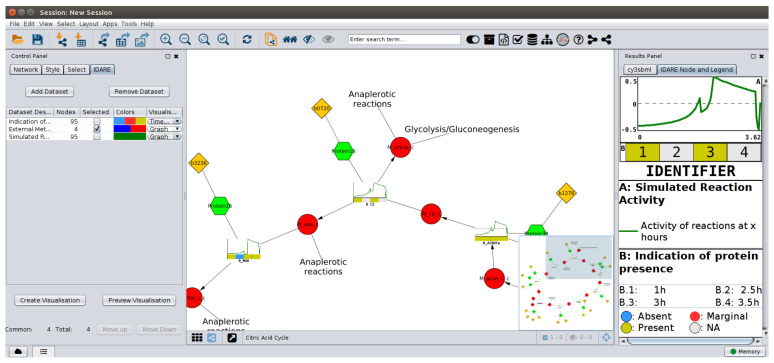
The interface of the app. On the left, the Data Management table provides an overview of available datasets and the currently selected properties for visualization. In the network window, the nodes with mapped images are displayed and on the right. The legend provides information about the visualization. Red circles represent metabolites, green hexagons proteins, and genes are represented as yellow rhombus. All reactions are replaced by metanodes in this figure.

**Figure 3 metabolites-11-00300-f003:**
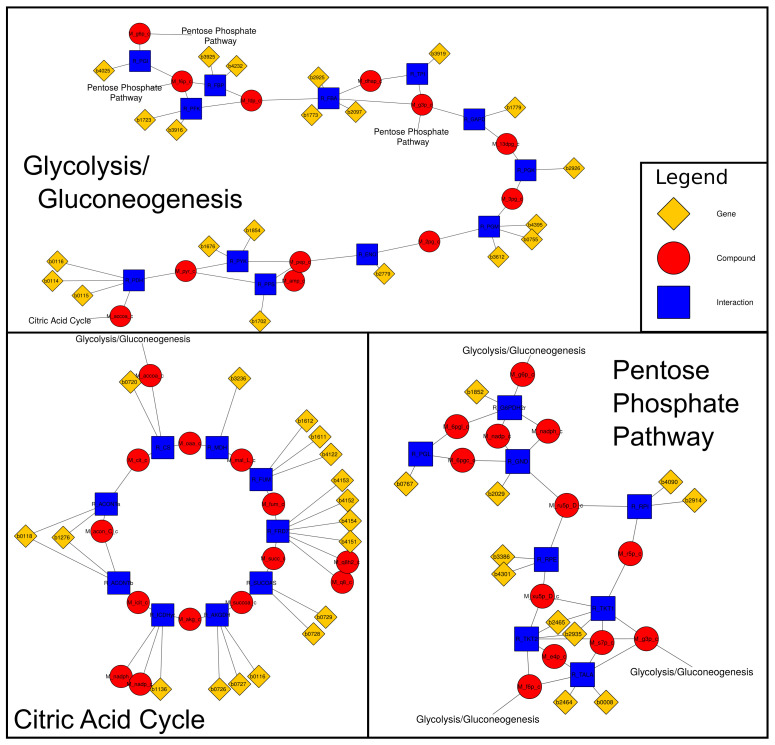
Connected subnetworks for glycolysis/gluconeogenesis, citric acid cycle and the pentose phosphate pathway. The pathways are linked to each other by the linkers (names of the other pathways). Layouts were generated automatically with Cytoscape layout algorithms and scaled/rotated to fit this figure. Gene nodes were added to the network based on the SBML note fields using IDARE2’s SBML annotation tool. Abundant metabolites like water and phosphate along with co-factors like ATP and Coenzyme-A were set to be removed from the network, while NADPH was set not to extend to other pathways.

**Figure 4 metabolites-11-00300-f004:**
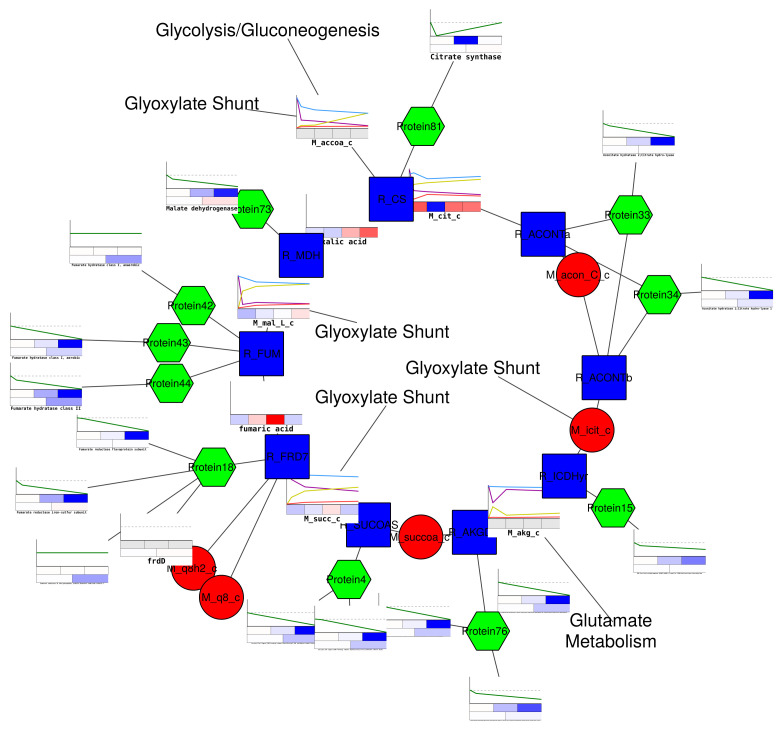
*E. coli* TCA (tricarboxylic acid cycle) with data mapped from multiple sources. Labeling data suggest usage of the glyoxylate shunt, as alphaketoglutarate is the only metabolite not accumulating labeling. Overall expression and protein levels are lower, but metabolite levels seem to increase upon treatment. Red circles represent metabolites, green hexagons represent proteins formed by the genes accoring to the GPR rules from the sbml. Blue squares represent reactions.

## Data Availability

All repositories and data used in this study can be found on our github pages at https://github.com/sysbiolux (accessed on 26 April 2021).

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
