# Peer review of "IDARE2—Simultaneous Visualisation of Multiomics Data in Cytoscape"

_metabolites, 2021, doi:10.3390/metabo11050300_

Round 1

Reviewer 1 Report

All concerns were properly assessed.

Reviewer 2 Report

The authors have revised their manuscript and provided useful documentation in GitHub for the user to follow the steps. The authors have also done a great job in answering all the questions.

Reviewer 3 Report

The authors have adequately addressed my comments. 

This manuscript is a resubmission of an earlier submission. The following is a list of the peer review reports and author responses from that submission.

Round 1

Reviewer 1 Report

In their manuscript, Pfau et al. present IDARE2, which is a Cytoscape app that allows for visualization of multi-omics data in the context of a metabolic network. The software is an update on IDARE, where major improvements are focused on less human intervention and more automated generation of maps.

In essence IDARE2 seems to be a valuable tool for multi-omics data analysis and visualization, and various aspects are cleverly addressed (e.g. splitting up into subnetworks). However, when reading one gets the feeling that this is an “old” manuscript. The latest cited reference is from 2016, and the sentence “more recent tools like Paintomics [2], BiNA [3] or MONGKIE [4]” from the Introduction makes one imagine that either these papers are from the last 1-2 years, or the previous paper (ref. [1]) is from the 90s, but they are all from 2010-2014, >7 years ago. Have there not been more recent efforts to address this issue? Please give some more recent perspective on the topic.

Another issue is that it is not entirely clear that the software being shown is version 2 of IDARE. The GitHub repository (https://github.com/sysbiolux/IDARE) only has a version 1 release, and also the Cytoscape App Store (http://apps.cytoscape.org/apps/idareidare) hosts version 1. Which version does one then use when installed from either source? Other links, such as http://idare-server.uni.lu/ (mentioned on GitHub), https://sysbiolux.uni.lu/IDARE and http://idare.uni.lu/ (mentioned in manuscript) are dead, further suggesting that this manuscript has been written a while ago.

When using the software, and following the tutorial on https://sysbiolux.github.io/IDARE-QuickStart/, not everything goes as it should (while using the model and data that the tutorial refers to). Step 1.5 throws some error from Cytoscape, although it actually doesn’t give any error text, so I cannot inform the authors what might be the issue. They might want to rerun their software on their own computer. In step 3.1, when selecting subnetworks, subsystems are only indicated by numbers and not names, which makes it very hard to see to which other subnetworks each subnetwork is connected. There might be more issues, I gave up already, but the authors should just run through their tutorial to make sure it works.

Otherwise, the manuscript is quite okay. Introduction would become more inviting when structured in a few paragraphs. The authors demonstrate its use on a small scale model, and mention in the discussion that they tested it in large models (25,000 edges), but it would be good to see results from integration with a genome-scale model. I gave it a try, but the previous errors made it very difficult to evaluate what was happening.

Reviewer 2 Report

In the present work the authors have generated a visualization platform that can overlay multi-omics data on reactions in metabolic networks and can be accessed through Cytoscape. The idea for the tool is of interest to the metabolic modelers who face challenge in representing multi-omics data in metabolic networks. Also, the availability of the application in Cytoscape makes it easier for the user to implement the tool. There are few suggestions and comments on the work.

  1. The authors need to restructure their methods and results sections. Figures 2 and 3 and their explanations are more relevant in results rather than methods section.
  2. The authors have represented the metabolites that are mostly part of the cytoplasmic compartment. How will the network look if the metabolites belong to different compartments such as extracellular and cytoplasm?
  3. The authors have considered the example of reactions and subsystems in E.coli. Is it feasible to implement the tool for metabolic networks for multi-compartment models of higher organisms such as humans and mouse? If yes, how does the tool handle the complexity of same metabolite in different subsystems and compartments?
  4. Does the tool work on only BIGG models or can it be used for models from other groups that do not follow the same nomenclatures for genes and metabolites? Can the user download the models in BioModels, KBase and other resources and visualize them using the tool?
  5. Which model of E.coli was used for the present analysis?
  6. How are the GPR associations handled in the representation? Figure 3 shows the genes represented as diamonds, but how will the user know if there is AND or OR association? Also, if transcriptomics data is integrated, how will the genes be handled? Will there be color difference based on the expression levels or size difference?
  7.  Escher (https://escher.github.io/#/) is also able to integrate transcriptomics and metabolomics data to metabolic models and the user is able to visualize the changes. Also, Escher has the added feature of carrying out FBA. Have the authors considered including such features in IDARE2?
  8. The link to IDARE2 (

    http://idare.uni.lu/IDAREDoc) is not directing to the correct page. Kindly check the URL.

Reviewer 3 Report

The authors describe a Cytoscape plugin for effective visualization of omics data from different sources in the context of metabolic (as well as other biological) networks. I recommend the acceptance of the paper after certain comments are addressed. 

IDARE presents an older version of the same tool. In the manuscript, IDARE is used instead of IDARE2 at certain places (e.g., line 109, line 154...). Please revise. 

Please provide a direct link to the IDARE2 repository in the "Extensibility" section. The last update of the IDARE repository was 2 years ago?

Genome-scale metabolic models (GEMs) and visualization of their simulation results are mentioned in the introduction. Can IDARE2 also be used to visualize the simulation results in a similar way as recent tools for the visualization of GEMs? See, e.g.
https://pubmed.ncbi.nlm.nih.gov/32442279/
https://pubmed.ncbi.nlm.nih.gov/22446067/
https://pubmed.ncbi.nlm.nih.gov/29461874/
https://pubmed.ncbi.nlm.nih.gov/32059585/
https://pubmed.ncbi.nlm.nih.gov/26313928/
It would be useful to be able to visualize the experimental data together with the simulation results. E.g., to observe the correlations between the activity of a specific gene in a given dataset and the simulated activity of the reactions that are regulated by the same gene.  

Line 154: "can" is missing.

Line 198: "it is" instead of "its".

Please explain the abbreviations at their first appearance, for example, LB, API, etc.

Please provide dynamic links in section 3.1. 
